# Functional, Structural and Proteomic Effects of Ageing in Resistance Arteries

**DOI:** 10.3390/ijms25052601

**Published:** 2024-02-23

**Authors:** Lars Jørn Jensen

**Affiliations:** Department of Veterinary and Animal Sciences, Faculty of Health and Medical Sciences, University of Copenhagen, DK-1870 Frederiksberg C, Denmark; lajj@sund.ku.dk

**Keywords:** ageing, resistance artery, myogenic tone, flow-mediated vasodilation, structural remodeling, hypertension, cardiovascular disease, neurodegenerative disease, proteomics

## Abstract

The normal ageing process affects resistance arteries, leading to various functional and structural changes. Systolic hypertension is a common occurrence in human ageing, and it is associated with large artery stiffening, heightened pulsatility, small artery remodeling, and damage to critical microvascular structures. Starting from young adulthood, a progressive elevation in the mean arterial pressure is evidenced by clinical and epidemiological data as well as findings from animal models. The myogenic response, a protective mechanism for the microcirculation, may face disruptions during ageing. The dysregulation of calcium entry channels (L-type, T-type, and TRP channels), dysfunction in intracellular calcium storage and extrusion mechanisms, altered expression of potassium channels, and a change in smooth muscle calcium sensitization may contribute to the age-related dysregulation of myogenic tone. Flow-mediated vasodilation, a hallmark of endothelial function, is compromised in ageing. This endothelial dysfunction is related to increased oxidative stress, lower nitric oxide bioavailability, and a low-grade inflammatory response, further exacerbating vascular dysfunction. Resistance artery remodeling in ageing emerges as a hypertrophic response of the vessel wall that is typically observed in conjunction with outward remodeling (in normotension), or as inward hypertrophic remodeling (in hypertension). The remodeling process involves oxidative stress, inflammation, reorganization of actin cytoskeletal components, and extracellular matrix fiber proteins. Reactive oxygen species (ROS) signaling and chronic low-grade inflammation play substantial roles in age-related vascular dysfunction. Due to its role in the regulation of vascular tone and structural proteins, the RhoA/Rho-kinase pathway is an important target in age-related vascular dysfunction and diseases. Understanding the intricate interplay of these factors is crucial for developing targeted interventions to mitigate the consequences of ageing on resistance arteries and enhance the overall vascular health.

## 1. Introduction

Ageing is associated with an increase in the occurrence of arterial hypertension and organ blood flow dysregulation, which are major risk factors for cardiovascular diseases, renal failure, retinopathy, stroke, and cognitive impairment in both sexes. Diameter regulation (vasoconstriction, vasodilation, and remodeling) of resistance arteries (small arteries and arterioles) plays a paramount role in the control of mean arterial blood pressure (MAP) and organ blood flow. Blood flow regulation is achieved as an interplay between neurohormonal regulation, metabolic regulation, myogenic tone/myogenic response (basal tone, autoregulation), flow-mediated vasodilation (+other endothelium-dependent dilating mechanisms), structural remodeling, and control of the capillary network (angiogenesis, rarefaction). In the present paper, the discussion of resistance artery ageing is limited to the roles of myogenic tone, flow-mediated vasodilation, and structural remodeling. The impact of proteomics and bioinformatics is discussed as a promising approach to obtain new insight into vascular ageing. The emphasis is on normal “healthy ageing” in resistance arteries, but it is inevitable to also provide examples of hypertension and other age-related diseases. A thorough understanding of the molecular mechanisms involved in the dysregulation of vascular tone and remodeling in ageing will provide a better basis for developing new targets and treatments for vascular complications involving the dysregulation of organ blood flow in the elderly population. Moreover, as sexual dimorphism in these responses may arise during the course of ageing, future studies will need to consider this.

## 2. Review Methodology and Structure

The cited papers were selected based on a simple search string on PubMed containing the relevant terms ageing, vascular, myogenic tone OR flow-mediated vasodilation OR structural remodeling. If needed, the search was narrowed down by including the term resistance artery and/or the organ of interest (e.g., brain or cerebral). The cited information was selected based on a direct comparison of relevant age groups with a clear presentation of data concerning the reviewed vascular functions. Papers were omitted if they merely presented data as “age-related” functions or diseases without a direct comparison between age groups. When presenting data, a consistent hierarchy of organism, functional integrity, body system, and vascular bed was employed. The assigned age groups were inspired by recent definitions [1,2]. Rats were divided into age groups as: young (3–8 months), middle-aged (12–18 months), or old (>22 months). Murine age groups were assigned as: young (2–5 months), mature (6–9 months), middle-aged (10–18 months), or old (>18 months). Biological pathway analyses (KEGG; REACTOME) were employed on proteins shown in Table 1 using DAVID Functional Annotation Tool (v2023q4) and Reactome Database (v87), respectively. STRING protein–protein interaction network (functional and physical interactions between individual proteins) was performed using the latest version of STRING database (v12.0).

## 3. Systolic Hypertension in Ageing

Arterial stiffening, which occurs in most humans during the ageing process, causes isolated systolic hypertension, and may progress to systolic–diastolic hypertension through endothelial dysfunction, enhanced contractility, and structural remodeling in resistance vessels [3]. The systolic hypertension observed with advanced age is distinct from the midlife hypertension with elevated diastolic and systolic blood pressures, which are due to the increased total peripheral resistance [4]. The increased pulsatility in large stiff arteries is augmented by an early systolic arrival of the reflected pressure wave due to the increased impedance in the microcirculation [3,5]. As the pressure wave is transmitted longer distally in the ageing systemic circulation, the microcirculation may be compromised. This is commonly observed in renal glomeruli, the blood–brain barrier (BBB), coronary microcirculation, and in retinal capillaries [6,7,8]. The types of microcirculatory damage incurred by the increased pressure pulsatility include decreased angiogenesis, capillary rarefaction, increased inflammation, BBB rupture, microhemorrhages, and, in case of the brain, white matter lesions [9,10]. Arterial stiffness in ageing is related to changes in extracellular matrix (ECM) properties with a loss of elasticity and remodeling of the elastin, collagen, and fibronectin content of the arterial wall [6,11]. Molecular determinants of large artery stiffening in ageing include the ECM proteins responsible for collagen breakdown and production, such as matrix metalloproteinases (MMPs), tissue inhibitors of MMPs (TIMPs), transforming growth factor-β1 (TGF-β1), and tissue transglutaminases (TGMs). In addition, proteins responsible for elastin or elastic fiber expression, coherence, and functional characteristics, such as fibrillins and fibulins [12,13], may be involved in age-related arterial stiffness. Increasing attention is given to smooth muscle phenotypic changes related to various stressors, such as mechano-transduction, oxidative stress, genetic, and epigenetic factors [14,15,16]. It is beyond the scope of this review to provide a complete list of the possible molecular determinants of large artery stiffening in ageing.

## 4. Increased Mean Arterial Pressure in Ageing

The large population-based Framingham clinical study tracking hemodynamic variables from young adulthood to old age showed that mean arterial pressure (MAP) in both men and women increases over their lifetimes, from early adulthood to ~60 years of age, and stays high albeit with a tendency to decline towards old age [17,18]. When exploring sex-dependent differences at the population level in four community-based US cohort studies, it became evident that MAP is higher in men than in women throughout their lifetime, but the elevation in MAP from baseline at young adulthood is significantly higher in females than in males [19]. An advantage of using mice for vascular ageing studies is that they age rather fast. Another important point is that 12–18-month-old normal (male) laboratory C57BL6 wildtype mice present with age-dependent hypertension [20,21]. Female mice were only hypertensive at 18 months of age [21], and estrogen-replacement therapy lowered MAP significantly to pre-hypertensive levels, whilst being dependent on a simultaneous increase in renal Angiotensin AT_2_-Receptor expression [22]. It should be noted that the tendency for rats and mice to develop hypertension with ageing may be associated with a high degree of inter-individual variability. The molecular determinants of age-dependent increases in MAP in humans have not been clarified, but are a subject of intense investigation due to the obvious advantage of targeting human hypertension in the elderly in a sex-dependent manner.

## 5. Myogenic Response Protects the Microcirculation, but May Be Disrupted in Ageing

The myogenic response is a smooth muscle-derived, mechano-activated vasoconstriction or vasodilatation to an increase or decrease in the intravascular pressure, respectively. This response is important for establishing basal (myogenic) tone (MT) in resistance vessels, and for blood flow autoregulation in the brain, kidney, and most other organs [23,24]. Human arteries: In human pial arteries of different calipers, the level of myogenic tone was not age-dependent [25], whereas in human posterior ciliary arteries of the eye, myogenic tone was inversely proportional to the age of the subject [26]. As the information about age-dependent changes in myogenic tone in human resistance vessels is scarce, we have to turn to data from rodent models. Coronary arteries: In coronary arterioles from Fischer 344 rats, myogenic constriction was reduced at old age in endothelium-intact preparations, but not in endothelium-denuded vessels [27]. Skeletal muscle arteries: In skeletal muscle arterioles of rat soleus and gastrocnemius muscles, myogenic tone was reduced at old age [28,29,30], whereas no age-dependent difference was detected in mouse superior epigastric arteries [31]. Cerebral arteries: In rat middle cerebral arteries, the endothelium seems to play a modulating role on myogenic tone in ageing. Myogenic tone was increased in old age in endothelium-intact preparations in old male [32] and female rats [33], but was significantly decreased in endothelium-denuded preparations from old male rats [32]. In mouse middle cerebral arteries, the myogenic tone was modestly reduced at old age in normotensive mice [34,35,36], while it was dramatically reduced in old compared to young mice with Ang-II induced hypertension [34,35]. In mouse parenchymal arterioles, the myogenic tone was increased in old age, but only in the presence of an intact endothelium [37]. These data suggest that in old age the endothelium of the cerebral vasculature releases a contractile factor (e.g., Endothelin-1, Thromboxane A_2_ or Superoxide) that enhances the spontaneous vascular tone, whereas “classical” myogenic tone derived by the mechanical activation of smooth muscle constriction is reduced in old age. Renal arterioles: In mice, the renal autoregulation was impaired in the kidney of old mice, due to impairments in the myogenic response and pressure-induced calcium increase in afferent arterioles [38]. In addition, the loss of the renal tubuloglomerular feedback response in old mice was associated with the down-regulation of A_1_-type adenosine receptors in afferent arterioles [38]. Mesenteric arteries: The myogenic tone in rat third-order mesenteric arteries was decreased in middle age [39]. This is confirmed by data from mouse second–third-order mesenteric arteries, in which middle-aged mice show a markedly decreased myogenic tone [40,41]. No sex-dependent myogenic tone differences were observed in mouse mesenteric arteries [40]. Interestingly, one study found an increased myogenic tone in mature adult mice (6–7 months) vs. a decrease in middle-aged mice (13–14 months) [41].

Thus, in 13 out of 15 studies, myogenic tone was decreased in middle-aged or old mice and rats. In cerebral and coronary vessels, the endothelium seems to play a modulating role by altering the myogenic tone in ageing. The disruption of myogenic reactivity may have deleterious effects on the integrity of the microcirculation and the organ function in middle age and old age. When the basal tone is absent or reduced, the vessels can no longer dilate and increase the perfusion to vasodilator stimuli (i.e., reduced flow reserve), causing a mismatch in demand and supply. Moreover, the autoregulation of blood flow is disrupted in critical organs, causing too little blood flow with a supply of oxygen and nutrients when the arterial pressure is low. Conversely, under high arterial pressures, the capillaries will face a mechanical disruption due to excessive microcirculatory pressure and flow. The disturbed blood flow regulation might trigger serious diseases such as stroke, neurodegenerative diseases, cardiac ischemia, renal failure, and retinopathy.

## 6. Molecular Determinants of Reduced Myogenic Tone in Ageing?

What are the molecular determinants of age-dependent alterations in myogenic tone in resistance vessels? Previous studies suggest that Ca^2+^ entry channels (L-type, T-type and TRP channels) and intracellular Ca^2+^ handling mechanisms (Ca^2+^ store release and extrusion) in vascular smooth muscle are dysregulated during ageing [1,2]. However, only limited information is available, making this subject an important area for future investigations.

### 6.1. Voltage-Gated Calcium Channels

L-type channels: In small mesenteric arteries from middle-aged mice, two studies from the same authors showed that smooth muscle mineralocorticoid receptor (MR) is the driver of the expression of a single micro-RNA, miR-155, which negatively regulates the expression of L-type channels (Ca_V_1.2) and AT_1_-receptors [42,43]. In middle-aged mice, a decreased expression of miR-155 was responsible for an increase in L-type channel expression and activity, increased AT_1_-R expression, and increased vasoconstriction [43]. These data explain the enhanced vasoconstrictor responses to Ang-II and hypertension in middle-aged mice, but do not account for the reduced myogenic tone with ageing. Other studies have shown a decrease in the L-type channel expression and function in mesenteric arteries from middle-aged rats [44] and cerebral arteries from old mice [45]. Interestingly, the reduced L-type channel expression in middle-aged rat mesenteric arteries was mirrored by the increased expression of miR-328, which negatively regulates the L-type channel expression [44]. Yet, other studies failed to find an age-dependent difference in L-type channel expression and function in small coronary arteries from old rats [46], or in small mesenteric arteries from mature adult [47] and old [48] mice. Due to the lack of consistency between the studies, changes in L-type channel expression and activity may not be the primary molecular determinant for decreased myogenic tone with ageing. T-type channels: Ca_V_3.1 T-type channels, which are responsible for myogenic tone in a low pressure range from 40 to 80 mm Hg [49], are dramatically downregulated by ageing [47]. However, the possible role of Ca_V_3.1 channels in the age-dependent reduction in myogenic tone remains to be determined. The Ca_V_3.2 T-type channel localized in smooth muscle plasma membrane caveolae suppresses myogenic tone in small mesenteric arteries from young mice by activating the SR Ca^2+^ release channel RYR2 and nearby BK_Ca_ K^+^ channels [1,47,50]. Although the Ca_V_3.2 expression in wildtype mice was unaffected by ageing, we found that this protective role of the Ca_V_3.2 channels in young animals against excessive myogenic tone vanished in mature adult and middle-aged mice, suggesting that a loss of Ca_V_3.2 channel function may play a pathophysiological role in age-related vascular dysfunction [47]. Subsequently, it was found that smooth muscle plasma membrane caveolae are disrupted by ageing, which can explain the loss of the Ca_V_3.2 channel-mediated activation of the RYR2/BK_Ca_ axis and suppression of myogenic tone in middle-aged mice [51]. A loss of BK_Ca_-mediated negative feedback on myogenic tone in ageing might also be due to a weaker physical coupling between SR/RYR2 complex and BK_Ca_ channels, for example via a reduced junctophilin/microtubular coupling between SR and the plasma membrane with less peripheral location of SR in aged arteries [1]. Finally, the administration for 4 months of the senolytic cocktail Dasatinib + Quercetin was able to rescue the Ca_V_3.2/RYR2/BK_Ca_-mediated suppression of myogenic tone in middle-aged mice [52]. Nevertheless, the loss of the Ca_V_3.2/SR/RYR2/BK_Ca_ axis by ageing does not explain the general age-dependent decrease in myogenic tone observed across most studies.

### 6.2. TRP Channels

The transient receptor potential cation channel subfamily C member 6 (TRPC6) is involved in myogenic tone development in rat cerebellar and posterior cerebral arteries [53]. We showed that the pressure-dependent activation of Phospholipase A_2_ activity and 20-HETE production causes an increase in agonist-activated TRPC6 currents, and this was critically involved in the myogenic response in rat small mesenteric arteries preconstricted with a low concentration of α_1_-agonist and neuropeptide Y [54]. Comparing Ang-II-induced hypertension in young vs. old mice, Toth and colleagues observed an increased TRPC6 expression in middle cerebral arteries of young hypertensive mice compared to young normotensive mice; however, this upregulation of TRPC6 channel expression was not observed in aged hypertensive mice [35]. Furthermore, the myogenic tone and pressure-induced increase in smooth muscle intracellular [Ca^2+^] in cerebral arteries were increased by Ang-II hypertension in young mice, but were decreased by more than 50% in old Ang-II hypertensive mice [34,35]. These studies suggest that the loss of TRPC6 channel upregulation in RAS-induced hypertension during ageing could be involved in the mechanism for the age-dependent disruption of myogenic tone.

### 6.3. Intracellular Calcium Handling Proteins

Due to a reduced plateau phase Ca^2+^ level after the caffeine/phenylephrine-induced SR Ca^2+^ store release in old murine mesenteric artery myocytes, it was suggested that “storage-operated” non-selective cation channels (TRPC; Orai channels) might be involved in an age-related disruption of myogenic tone [48]. Moreover, a reduction in the expression of intracellular Ca^2+^ handling proteins (Ca_V_1.2, RYR2, SERCA2, PLB, and STIM1) caused a global decrease in the amplitude of Ca^2+^ signals, and a reduction in the rate of Ca^2+^ store refilling [45]. These changes might also be a contributing factor to the reduced myogenic tone in aged arteries. Mice with a missense mutation in Collagen IV alpha 1 (Col4a1), associated with small vessel disease-like brain pathology, had a reduced myogenic tone and pressure-induced depolarization by middle age compared to young mutant mice and age-matched control mice [55]. Furthermore, the middle-aged Col4a1 mutant mice had a disrupted SR Ca^2+^ release and reduced IP_3_-receptor mediated, Ca^2+^-dependent TRPM4 channel activation, which can explain the disruption of the pressure-induced smooth muscle cell depolarization and myogenic tone development in the middle-aged mutant mice [55]. Thus, the disruption of the SR Ca^2+^ store release may be involved in the reduced myogenic tone development in patients with cerebral small vessels disease and/or vascular dementia.

### 6.4. Potassium Channels

BK_Ca_ channels: BK_Ca_ channel expression and function were decreased in coronary arteries from old human subjects and old rats [56]. Likewise, the BK_Ca_ channel expression in soleus muscle arterioles was decreased in old rats [29]. In middle cerebral arteries, BK_Ca_ channels were upregulated in old male rats and downregulated in old female rats, leading to opposite changes in myogenic tone [33]. In mouse mesenteric arteries, the coupling between Ca_V_3.2 T-type channels and BK_Ca_-mediated negative feedback on myogenic tone was lost in middle-aged mice compared to young mice [47]. Furthermore, the BK_Ca_ channel blocker iberiotoxin did not eliminate the age-dependent reduction in myogenic tone in cerebral arteries from aged hypertensive mice [34]. K_V_ channels: The K_V_ channel blocker 4-aminopyridine increased the myogenic tone from a reduced level in aged rat skeletal muscle arterioles to the same level as in young arterioles [29]. Kir2x channels: A recent study suggested that Kir2.x channels are upregulated in smooth muscle cells of cerebral parenchymal arterioles in old mice [37]. Whereas the decreased BK_Ca_ channel expression and function cannot explain the disruption of myogenic tone in ageing, an increased activity or expression of smooth muscle BK_Ca_ (male rats), K_V_, and Kir2.x channels can. More data on the function and expression of K_V_ and Kir2.x channels in resistance vessels with myogenic tone will be necessary to elucidate their role in ageing.

### 6.5. RhoA/Rho-Kinase Pathway

In addition to the Ca^2+^ signaling pathways, we aimed to investigate the possible age-dependent role of altered smooth muscle contractile Ca^2+^ sensitivity. Phosphorylation of the regulatory subunit of Myosin Light Chain is a balance between the activities of the Ca^2+^/Calmodulin-activated Myosin Light Chain Kinase and the Myosin Light Chain Phosphatase, the latter being negatively regulated by phosphorylation via Rho-kinase. Initially, we found that the expression of Rho-kinase 2 (ROCK2) was increased in mature adult mice compared to young mice, and that the ROCK2 inhibitor KD025 elicited a stronger inhibition of myogenic tone in the mature adult mice [41]. The increased myogenic tone in mature adult mice was paralleled by our observation of the increased noradrenaline-induced tone in this age group [47]. Whereas these data could only explain the increase in myogenic tone in mature adult mice, we needed more data to explain the decline in myogenic tone in middle age and old age. This prompted us to investigate the proteomic effects of ageing in middle cerebral arteries (MCA) and second–third order mesenteric arteries (MRA) by comparing 8 young (3 months old) and 7 middle aged (14 months old) mice using mass spectrometry-based proteomics [57]. In total, 207 proteins were significantly differentially expressed and/or hierarchically clustered in an age-dependent manner, corresponding to ~11% of the uniquely identified and quantified proteins [57]. In our KEGG biological pathway analysis, we discovered that the regulation of the actin cytoskeleton pathway was one of the top significantly enriched pathways. Central in this pathway is the RhoA/Rho-kinase pathway, confirming our initial hypothesis that this pathway was important for age-dependent changes in vascular function. Several proteins connected with this pathway were on the list of age-dependent proteomic changes: ARHGEF7, PAK2, ROCK1, M-RIP, ML12B, MYO5B, and ACTB [57]. Work is underway to try to understand whether any of these changes are underlying the changes in myogenic tone with ageing, but in fact, the down-regulation of ROCK1 and ML12B could explain this alone.

Wnt3a of the canonical Wnt pathway is activated during advanced arterial ageing and is associated with vascular injury [58,59]. Furthermore, the non-canonical Wnt pathway member Wnt5a, capable of activating RhoA, is involved in the excess vasoconstriction and salt-sensitive hypertension in middle-aged mice, a phenotype that was rescued by supplementation with the anti-ageing factor Klotho [60]. Interestingly, Wnt10a of the canonical Wnt/β-catenin pathway was significantly upregulated in the proteome of middle cerebral and small mesenteric arteries from middle-aged mice [57]. There is a paucity of data concerning the role of Wnt signaling in myogenic tone, but in human bronchi, mechanical stretch caused an increase in the expression of a large number of genes in the Wnt-mediated signaling pathway [61]. It is tempting to speculate that some components or mediators of the canonical or non-canonical Wnt pathways might be involved in the disruption of myogenic tone in ageing.

## 7. Flow-Mediated Vasodilation—Indicator of Endothelial (Dys-)Function in Ageing

Flow-mediated vasodilation (FMVD) is an endothelium-dependent, shear stress-activated vasodilation found in large and small arteries [47,62,63]. The FMVD responses are regarded as the hallmark of endothelial function, which is directly correlated with cardiovascular health [64]. Large human arteries: Non-invasive FMVD measurements in humans are routinely measured on brachial arteries using the post-occlusion increase in blood flow to calculate the percent increase in FMVD [62,65]. In children and adolescents, there was a trend to an age-dependent decline in FMVD, but there were no significant sex-differences [66]. Conversely, FMVD in large conduit arteries was markedly reduced in old subjects compared to young adults [67,68,69,70,71,72]. In a large, community-based study, FMVD in both sexes was negatively correlated with age, with men having an overall lower FMVD than women, but women having a steeper age-dependent decline in FMVD [73]. In brachial arteries from young and aged otherwise healthy subjects, FMVD correlated negatively with the arterial endothelial expression of ROS-derived nitrotyrosine [70]. In aged subjects, the expression of mitochondrial ROS-inducing NAD(P)H oxidase and the inflammation-induced transcription factor NFκB was increased in the arteries, pointing to a role of oxidative stress and vascular low-grade inflammation in the reduced FMVD in ageing [70]. Supplementation with MitoQ to quench the mitochondrial superoxide production improved the FMVD in the brachial artery in old (60–79 years), otherwise healthy, subjects [74]. Human arterioles: In human coronary arterioles, there was an age-dependent transition in the mediator of FMVD from prostacyclin (PGI_2_) in children to NO in young (18–55 years) and older (>55 years) subjects without coronary artery disease, whereas in older patients with coronary artery disease, the remaining FMVD was mediated via mitochondrial H_2_O_2_ production [75]. H_2_O_2_ released from the endothelium acts as a vasodilator via the activation of smooth muscle K^+^ channels, thereby eliciting Endothelium-Dependent Hyperpolarization (EDH) and vasodilation [76,77]. EDH is defined as the ability of an endothelium-dependent vasodilator to cause dilation via the hyperpolarization of smooth muscle cells derived by a factor released from the adjacent endothelial cells (K^+^; H_2_O_2_; endothelial hyperpolarization spreading to smooth muscle via myoendothelial junctions) [78]. In resistance arteries, it is widely accepted that EDH functions as a compensating vasodilator mechanism when the NO-mediated vasodilation is reduced [79].

Rodents: Whereas in humans FMVD is primarily a measure of large (brachial) artery endothelial function, in rodent models FMVD can be measured in resistance arteries and arterioles using ex vivo pressure myography. Coronary arterioles: In rat coronary arterioles, ageing reduced FMVD and flow-induced NO and H_2_O_2_ production, which were linked with an increased superoxide production in coronary arterioles from old rats [80]. In old female rats, estrogen replacement therapy could improve the age-dependent loss of NO bioavailability and FMVD in coronary arterioles [81]. The treatment with stromal vascular fraction cells or adipose-derived stem cells was able to reverse oxidative stress and rescue the age-dependent decline in FMVD in coronary arterioles from aged female rats [82]. Overall, there was a gradual shift from FMVD being dependent on NO production at a young age towards being dependent on the vasodilator H_2_O_2_ (acting as EDH) at old age in rat coronary arterioles [80,81]. Skeletal muscle arteries: FMVD measured ex vivo in soleus muscle arterioles was reduced by ~50% in old male compared to young rats, and this difference was eliminated after the partial blockade of NO production using the nitric oxide synthase (NOS) inhibitor L-NAME [83]. In superior epigastric arteries from old mice, the age-dependent decline in NO-mediated vasodilation was compensated for by an increase in the mitochondrial Ca^2+^-release and activation of SK/IK Ca^2+^-activated K^+^ channels acting as EDH [84]. Thus, the endothelium may, in some skeletal muscle arteries, be resilient to the age-induced loss of NO production through the upregulation of EDH. FMVD in mouse femoral arteries was reduced significantly in old mice, but this decline was rescued by activating endothelial P2Y_1_-receptors using 2-Me-ATP [85]. FMVD measured using ultrasonography in femoral arteries in mice was significantly reduced in old compared to young mice, showing that it is possible to assess FMVD in vivo in murine arteries [86]. Mesenteric arteries: In small mesenteric arteries from middle-aged rats, FMVD was reduced as compared to young rats, and this age-dependent decline was improved by TNFα blockade [87]. There was a shift from a flow-induced rise in NO production in young rats to a decreased NO production in middle-aged rats with an increased NAD(P)H oxidase-derived H_2_O_2_ production [88]. In small mesenteric arteries from middle-aged rats, there were no sex-dependent differences in the maximal total FMVD, but there was a shift from NO-mediated to EDH-mediated vasodilation in middle-aged female rats [89]. In small mesenteric arteries from old rats, FMVD was reduced compared to young rats [90], and the decrease in the shear stress-induced release of NO was linked to an increased production of ROS, a reduced activity of SuperOxide Dismutase (SOD), and an impaired shear stress-induced activation of eNOS [91]. In mouse small mesenteric arteries, FMVD was reduced by ~50% in young mice deficient in the T-type Ca^2+^ channel Ca_V_3.2, whereas no difference was observed in mature adult and middle-aged Ca_V_3.2KO vs. wildtype mice, suggesting the loss of a protective role of endothelial Ca_V_3.2 channels in ageing.

To summarize, all studies from human and rodents in all vascular beds so far examined (large conduit arteries and resistance vessels) point to an impairment of FMVD at middle age and old age. An abundance of studies point to a role of excess oxidative stress and a low-grade inflammatory response, involving a reduction in NO bioavailability and a partial compensation by the increased role of EDH, such as H_2_O_2_. The molecular determinants of an age-dependent FMVD decline are thus likely to include changes in the expression and function of NO, H_2_O_2_, Superoxide, eNOS, NAD(P)H oxidase, SOD, catalase, TNFα, and/or NFκB. The deficiency of protective mechanisms, such as estrogen in females, and endothelial Ca_V_3.2 T-type channels or P2Y_1_-receptors may also play a role.

## 8. Resistance Artery Remodeling—A Hypertrophic Response to Ageing

The structural remodeling of resistance arteries, causing either an inward or outward change of the maximal lumen diameter and a hyper- or hypotrophic change in the vascular wall material, is a chronic adaptation to long-term changes in pressure and/or flow as well as disease states [92,93]. The relevant structural/dimensional parameters to consider in vascular remodeling are: Lumen diameter; media:lumen-ratio (alternatively, wall/lumen-ratio); cross-sectional area of vessel wall, distensibility, and/or incremental elastic modulus (slope or β-value of stress/strain curve). In terms of clinical relevance, an increased wall/lumen-ratio is an independent risk factor for serious cardiovascular events [94,95]. Typical remodeling patterns are: (1) a persistent increase in intraluminal pressure (i.e., hypertension) leads to inward remodeling without hypertrophia (inward eutrophic remodeling; increased media:lumen-ratio), which normalizes wall stress. (2) Persistent increase in intraluminal flow (i.e., normal growth, pregnancy, physical activity) will cause an outward hypertrophic remodeling, which normalizes shear stress. (3) A persistent stimulation by vascular growth factors (e.g., under obesity, diabetes) will cause a hypertrophic growth in the vascular wall, which can be followed either by an outward remodeling (normalizes wall stress) or an inward remodeling (large increase in wall stress). (4) Persistent anti-hypertensive treatment with a vasodilator will cause an outward remodeling. Since ageing causes long-term changes in blood pressure, myogenic tone, flow-mediated vasodilation, and possibly in other vasomotor responses, it follows that ageing should elicit structural remodeling of resistance vessels.

Large arteries—human and rodent studies: In ageing, a consistent enlargement of the aortic media:lumen-ratio (M/L-ratio) is noted in humans and rodent models [96,97,98]. In human renal arteries, the media layer showed outward hypertrophic remodeling, whereas the intima layer showed inward hypertrophic remodeling with ageing [99]. The expression of Matrix Metallo-Proteinase 2 (MMP2) and MMP9 are increased in an age-dependent manner in aortae of rats, non-human primates, and humans [100,101,102]. MMP2 activation leads to a proteolysis of extracellular matrix proteins, such as collagen and elastin, thus enabling the remodeling process. However, in old rat aortae, MMP2 also increases the expression of the growth factor TGFβ_1_ and activates the TGFBR2-receptor signaling, which leads to an increase in fibronectin and collagen expression, and promotes fibrosis [103]. Vascular stiffness and pulse wave velocity correlated significantly with plasma MMP9 levels in patients with isolated systolic hypertension, suggesting MMP9 as an age-related plasma biomarker [104,105]. This is supported by a study showing that plasma levels of MMP9 were increased in old male and female mice [106]. In a rat model of isolated systolic hypertension, arterial stiffening was associated with the aortic activity of transglutaminase 2 (TGM2) [107]. TGM2 is secreted into the interstitial space and functions to crosslink extracellular matrix proteins, such as collagen, to complete the remodeling process [93]. Resistance arteries—humans: In human subcutaneous resistance arteries, there was a positive correlation between M/L-ratio and age in both normotensive and hypertensive subjects irrespective of sex, but with a steeper age-dependent increase in hypertensives [6,108].

Rodent studies—Coronary arterioles: In coronary arterioles from old rats, the wall:lumen-ratio (W/L-ratio) was increased while the elastic modulus and vessel stiffness were decreased [109]. Skeletal muscle arterioles: In old rats, skeletal muscle feed arteries and arterioles displayed outward hypertrophic remodeling with no change in W/L-ratio [110]. Cerebral arteries: Hypertrophic remodeling with an increased W/L-ratio (but no change of lumen diameter) was observed in basilar arteries from old rats [111,112]. Inward hypertrophic remodeling (a large increase in W/L-ratio) with no change in distensibility was found in posterior cerebral arteries from mature rats (11 months) [113]. Outward remodeling was observed in five different intracranial arteries (BA, ACA, MCA, ICA, and PCA) from old mice [114]. Likewise, in old mice, posterior cerebral arteries and parenchymal arterioles showed outward eutrophic remodeling [115]. Genital and uterine arteries: Inward hypertrophic remodeling was shown for pudential arteries from old male Spontaneously Hypertensive Rats (SHR) with erectile dysfunction compared to young SHR [116]. This inward hypertrophic remodeling was presumably linked to the hypertension in the SHR model. An outward hypertrophic remodeling and increased vessel stiffness were observed in uterine arteries from old female mice [117].

Mesenteric arteries: An outward hypertrophic remodeling with no change in M/L-ratio was observed in mesenteric small arteries from normotensive old (>8 years) vs. young (1.5–2 years) sheep [118]. Small mesenteric arteries in old rats displayed outward hypertrophic remodeling with an increased thickness of the media layer and unchanged adventitial layer [119]. However, the number of smooth muscle cells in the media layer was unchanged [119]. The capacitance and circumference of mesenteric artery myocytes were increased in old mice [48], supporting the idea that hypertrophic remodeling is due to increased vascular smooth muscle cell size in ageing.

The high flow-induced outward remodeling observed in small mesenteric arteries from young rats was absent in old rats, but the flow-induced hypertrophic remodeling was similar in both age groups [120]. This indicates an increased W/L-ratio under high-flow conditions in old rats. Moreover, mesenteric arteries released more Superoxide and showed a higher expression of Superoxide Dismutase in old rats than in young rats [120]. With no change of eNOS expression in the old rats, these data suggest a ROS-induced decrease in NO bioavailability as the mechanism for the lack of flow-induced outward remodeling in old rats [120]. To this end, a selective agonism of the anti-inflammatory Angiotensin AT_2_-receptor restored the lack of flow-induced outward remodeling in small mesenteric arteries from old mice [121].

Outward hypertrophic remodeling, lowered distensibility, and an increased elastic modulus (β-value) was observed in small mesenteric arteries from middle-aged mice [122]. In the same study, middle-aged mice deficient in Caveolin-1 (Cav-1) showed outward hypotrophic remodeling and lowered distensibility, suggesting that caveolae and Cav-1 are required for the hypertrophic remodeling [122]. However, the hypertrophic remodeling observed in small mesenteric arteries from mature adult and middle-aged mice was unaffected by the knockout of Ca_V_3.2 T-type Ca^2+^ channels [47], which are concentrated in caveolae in wildtype mice.

Sex-dependency: In old female rats, but not in male rats, the flow-induced outward remodeling response in mesenteric small arteries was preserved [123]. In the mouse model of transgenic familial Alzheimer’s Disease (3xFAD), middle-aged female mice, but not male, showed an outward hypertrophic remodeling with an increased M/L-ratio in small mesenteric arteries compared to non-transgenic control mice [124]. This could be related to the fact that females are more susceptible to Alzheimer’s Disease than males, but the latter two studies also suggest the role of female sex hormones in structural remodeling. More studies are needed to unravel the sex-dependency of structural remodeling in ageing.

To summarize, most of the studies suggest that outward hypertrophic remodeling is the common change in ageing, with the alternative possibility that an inward hypertrophic remodeling is linked with hypertension or other comorbidities. A majority of studies reporting such data point to an increased stiffness of resistance arteries with ageing. Sex-dependent changes are observed in aged arteries, and it will be important to unravel the exact nature and mechanisms of these changes in future studies.

## 9. Potential Molecular Mechanisms in Age-Dependent Structural Remodeling

From the studies above, it is clear that reduced NO signaling/bioavailability and increased oxidative stress is a key component in age-dependent remodeling responses. The oxidative stress activates gelatinases (MMP2; MMP9) that digest ECM fibers and activate TGFβ1 signaling, which via TGFBR2 receptors promotes growth and/or fibrosis in ageing. Transglutaminases (TGM2) may be involved in the crosslinking of the degraded protein fibers. In addition, caveolae containing Cav-1 seem to play a role in age-dependent remodeling, which is not surprising given the many receptors and signaling molecules that are concentrated in caveolae. Estrogen receptors and a reduced estrogen production associated with menopause likely plays a role in the sex-dependency of remodeling observed with ageing. The involvement of receptors for angiotensin II, aldosterone, prostaglandins, thromboxane, estrogen, and endothelin-1 in age-dependent structural remodeling and arterial stiffening has been extensively reviewed elsewhere [3,125,126,127,128]. It is also clear that pro-inflammatory signaling and so-called “sterile” inflammation plays an important role in the age-associated vascular remodeling process [3].

Thrombospondin-1 (TSP-1) is involved in structural remodeling in large and small arteries [129,130]. Physiological levels of TSP-1 increased Superoxide production and decreased NO-dependent vasodilation more in coronary arterioles from old female rats than in young females [131]. The vascular remodeling index after hindlimb ischemia was significantly lower in old compared to young wildtype mice. However, in TSP-1-deficient mice, the remodeling index during ischemia was increased in both young and old mice, and the age-dependent difference was eliminated [132]. These data confirm a role for TSP-1 to suppress remodeling responses in ageing. In an investigation of age-dependent changes in the proteome of murine and human left ventricular cardiomyocytes, lactadherin (MFGE8) protein was significantly increased by ageing [133]. In coronary arteries, lactadherin accumulation was observed in the interstitial space between endothelial and smooth muscle cells [133]. Coronary artery endothelial cells cultured on immobilized lactadherin showed an increased expression of the pro-inflammatory cytokine IL6 and the activation of age-related signaling cascades [133]. Future studies should investigate whether the increased lactadherin expression in aged resistance vessels promotes inflammatory signaling and structural remodeling in an age-dependent manner.

Our investigation of age-dependent changes of the proteome in middle cerebral arteries and small mesenteric arteries in middle-aged mice [57] showed significant up- or down-regulation of the following ECM or cytoskeletal structural components in aged arteries: MFAP2, LAMA3, FBLN4, and TBB1. Furthermore, the proteins ZCHC3 and KV6AB involved in immune system activation or signaling were significantly upregulated, and the oxidases AOXA and COX5B were downregulated in aged arteries [57]. The enriched KEGG biological pathways with potential relevance to vascular structure and remodeling were: Vitamin B6 metabolism, Biosynthesis of antibiotics, Regulation of actin cytoskeleton, Endocytosis, Focal adhesion, and ECM-receptor interaction [57]. These data highlight the potential usefulness of mass spec-based proteomics in elucidating known as well as hitherto unknown mechanisms and pathways involved in age-dependent changes in resistance arterial structure and function.

## 10. Protein–Protein Interactions and Pathway Analyses of Age-Dependent Mechanisms

Proteins with a potential age-dependent role mentioned in this review are listed in Table 1. Based on this list, pathway enrichment analyses are shown in Table 2 (KEGG biological pathways) and Table 3 (REACTOME pathways). A protein–protein interaction (STRING) network is shown in Figure 1. Furthermore, the STRING analysis yielded a list of predicted functional partners of the proteins in the network, as shown in Table 4.

The list of age-dependent proteins in Table 1 should serve as inspiration for future work regarding the role of specific proteins in the age-dependent decline in resistance artery function. Most frequent among the 15 most highly enriched KEGG pathways are cellular signaling pathways, such as calcium signaling, cGMP-PKG signaling, Relaxin signaling, Advanced Glycation Endproducts (AGE) and receptor for AGE (RAGE) signaling, Focal adhesion pathway, and Oxytocin signaling. The second most enriched KEGG pathways are brain disease pathways, where Neurodegenerative Diseases and Alzheimer’s Disease pathways appear, stressing the fact that ageing is the biggest risk factor for neurodegenerative disorders. Other disease pathways are cancer-related, diabetic complications, dilated cardiomyopathy, and a hormonal disorder (Cushing’s Syndrome). Among the top 15 REACTOME pathways enriched, 7 of them are cellular signaling pathways, and 6 denote physiological functions (contraction, homeostasis, conduction, and hemostasis). Overall, cellular signaling pathways are frequently affected by ageing in resistance arteries, and disease pathways are well represented, indicating that ageing confers a high risk of developing these diseases. However, processes involved in intercellular communication, ECM–cell interaction and cell–cell adhesion processes, such as electrical conduction, focal adhesion, and EPH-Ephrin signaling also seem to play a role in the ageing vasculature. In the STRING interaction network (Figure 1), the most busy interaction cluster consists of regulators of structural proteins, showing close interactions between MMP9, MMP2, TGFβ1, TNFα, and Decorin. Decorin, which is a regulator of fibril formation, interacting with several other ECM proteins, was predicted by the STRING database as a functional interaction partner with proteins in Table 1. Another small interaction cluster is centered on the L-type voltage-gated Ca^2+^ channels (Ca_V_1.2), with intracellular Ca^2+^ handling proteins, T-type channels, TRP channels, and K^+^ channels also belonging to this cluster. The G-protein coupled receptors (*AGTR1*, *AGTR2*, *EDNRA*, *EDNRB*, *TBXA2R*, *PTGFR*, *P2YR1*, and *ADORA1*) form a similar cluster aside from the main network, with several connections to the other clusters. The actin cytoskeleton regulatory proteins *ROCK1*, *ROCK2*, *MYL12B*, *PAK2*, and *ARHGEF7* form an additional cluster at the side of the largest cluster harboring the structural matrix-regulating proteins. Overall, these pathways and interactions point to central roles of intracellular calcium, extracellular matrix structural proteins, actin-cytoskeleton regulators, and G protein-coupled receptors as being major molecular targets for understanding and targeting age-dependent changes in resistance artery structure and function. In order to solve the complex role of ageing in resistance arteries, more mass-spec based proteomics studies are needed, including special applications for posttranslational modifications and membrane proteins.

## 11. Discussion and Concluding Remarks

The STRING analysis shown in Figure 1 enables an overview of the physical and functional interactions between the proteins with a suggested age-dependent role. The same analysis also led to a suggestion of additional interacting protein partners (see Table 4) to the existing protein–protein interaction network. For example, a literature search indicated that the proteins Decorin, NFKFBIA, RelB, TGFB3, TIMP1, TNFRSF1A, and UQCRC1 might be involved in vascular ageing.

Figure 2 presents an overview of the changes found regarding myogenic tone, flow-mediated vasodilation, and structural remodeling in ageing resistance arteries. With a preserved endothelial function and FMVD, the decline in myogenic tone would tend to cause vasodilation. Myogenic tone, per se, is not a direct cause of changes in blood pressure, but adjusts the basal tone and autoregulation of blood flow around a set point at the prevailing blood pressure. Therefore, with a normal blood pressure (normotension), a chronic loss of myogenic tone would tend to cause outward remodeling [134,135] (see upper part of Figure 2). By simultaneously stimulating a vessel wall growth resulting in outward hypertrophic remodeling, the aged artery under normotension will maintain an unchanged M/L-ratio. This remodeling response is reminiscent of the physiological remodeling occurring with increased blood flow and a general increase in cardiac output (pregnancy, normal growth, physical activity, obesity, etc.). Conversely, if the predominant change in ageing is a disruption of endothelial function and FMVD with little change in myogenic tone, the vessel will undergo acute vasoconstriction, as shown in the lower part of Figure 2. Endothelial dysfunction often leads to hypertension, and with prolonged vasoconstriction due to the disruption of FMVD and an increased wall stress due to hypertension, the aged vessel will respond by inward hypertrophic remodeling (see lower part of Figure 2). Figure 2 summarizes the major mechanisms that might be involved in the disruption of myogenic tone, FMVD, and the changes in arterial structure. The reader is referred to Table 1, Table 2, Table 3 and Table 4 for more insight into the molecular mechanisms and pathways that become activated or deficient by ageing. Finally, more proteomic and bioinformatics studies are needed to advance our understanding of the age-dependent changes in resistance artery structure and function.

This review focused on age-dependent changes in myogenic tone, flow-mediated vasodilation, and structural remodeling. Other mechanisms, such as local metabolic regulation, neuro-hormonal regulation, capillary rarefaction, or angiogenesis, might also be involved in age-dependent changes in resistance artery structure and function. It is my hope that this review will stimulate more work and new approaches in the study of the effects and possible therapeutic management options of an age-dependent decline in resistance artery function.

## Figures and Tables

**Figure 1 ijms-25-02601-f001:**
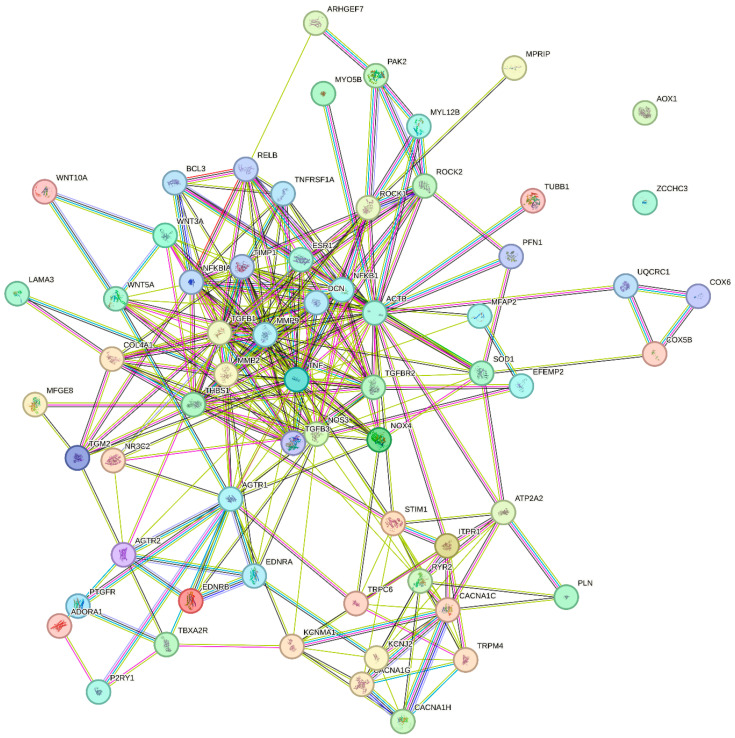
Protein–protein interaction network. The proteins in Table 1 were analyzed using STRING database to show their respective physical and functional interactions with medium confidence level (Score ≥ 0.400). See Table for definition of symbols.

**Figure 2 ijms-25-02601-f002:**
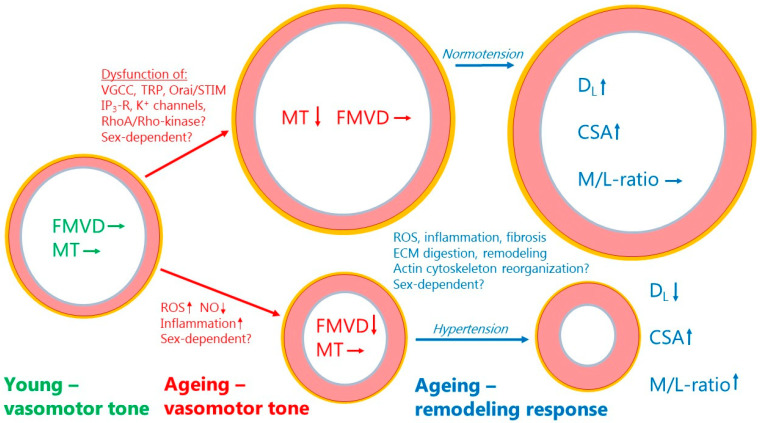
Summary of age-dependent changes in myogenic tone, flow-mediated vasodilation, and structural remodeling in resistance arteries. Green text and arrows to the left denote vasomotor tone at young age with a normal MT and FMVD. Red text and arrows in the middle denote changes to vasomotor tone during ageing. Upper part denotes decreased myogenic tone but no change in FMVD leading to vasodilation. Lower part denotes decreased FMVD but no change in MT leading to vasoconstriction. Blue text and arrows to the right denote chronic changes due to remodeling during ageing. Upper part shows the final remodeling outcome with a normal blood pressure, and lower part shows the final remodeling outcome in case of co-existing hypertension. See text for further explanations. The inner endothelial (intima) layer is shown as gray-blue, the smooth muscle (media) layer is shown as pink, and the connective tissue (adventitia) layer is shown in yellow color. Arrows denote the direction of change in a function, factor, or parameter. MT (myogenic tone); FMVD (flow-mediated vasodilation); D_L_ (passive lumen diameter); CSA (cross-sectional area); M/L-ratio (media:lumen-ratio); ROS (reactive oxygen species); NO (nitric oxide); VGCC (voltage-gated calcium channel); TRP (transient receptor potential channel).

**Table 1 ijms-25-02601-t001:** List of proteins with possible age-dependent role in resistance arteries (see text).

Uniprot ID (Homo Sapiens)	Gene Name	Protein (Short Name)	Protein Name
P60709	ACTB	Beta-actin	Actin, cytoplasmic 1
P30542	ADORA1	A_1_-R	Adenosine receptor A1
P30556	AGTR1	AT_1_-R	Type-1 angiotensin II receptor
P50052	AGTR2	AT_2_-R	Type-2 angiotensin II receptor
Q06278	AOX1	AOXA	Aldehyde oxidase
Q14155	ARHGEF7	RhoGEF7/β-Pix	Rho guanine nucleotide exchange factor 7
P16615	ATP2A2	SERCA2	Sarcoplasmic/endoplasmic reticulum calcium ATPase 2
O73707	Cacna1C	Ca_V_1.2	Voltage-dependent L-type calcium channel subunit alpha-1C
O43497	Cacna1g	Ca_V_3.1	Voltage-dependent T-type calcium channel subunit alpha-3G
O95180	Cacna1h	Ca_V_3.2	Voltage-dependent T-type calcium channel subunit alpha-3H
P02462	COL4A1	Col4a1	Collagen alpha-1(IV) chain
P10606	COX5B	COX5B	Cytochrome c oxidase subunit 5B, mitochondrial
P25101	EDNRA	ET_A_-R	Endothelin receptor type A
P24530	EDNRB	ET_B_-R	Endothelin receptor type B
P03372	ESR1	ER	Estrogen receptor
O95967	FBLN4	Fibulin-4	EGF-containing fibulin-like extracellular matrix protein 2
Q14643	ITPR1	InsP_3_R1	Inositol 1,4,5-trisphosphate receptor type 1
P63252	KCNJ2	Kir2.1	Inward rectifier potassium channel 2
Q12791	KCNMA1	BK_Ca_ α-subunit	Calcium-activated potassium channel subunit alpha-1
Q16787	LAMA3	Laminin α3	Laminin subunit alpha-3
P55001	MFAP2	MFAP-2	Microfibrillar-associated protein 2
Q08431	MFGE8	MFG-E8	Lactadherin
P08253	MMP2	MMP-2	Matrix metalloproteinase-2
P14780	MMP9	MMP-9	Matrix metalloproteinase-9
Q6WCQ1	MPRIP	M-RIP/p116^RIP^	Myosin phosphatase Rho-interacting protein
O14950	MYL12B	MLC_20_	Myosin regulatory light chain 12B
Q9ULV0	MYO5B	MYO5B	Unconventional myosin-Vb
P19838	NFKB1	NF-κB	Nuclear factor NF-kappa-B p105 subunit
P29474	NOS3	NOS3/eNOS	Nitric oxide synthase 3
Q9NPH5	NOX4	NOX4	NADPH oxidase 4
P08235	NR3C2	MR	Mineralocorticoid receptor
P47900	P2RY1	P2Y_1_-R	P2Y purinoceptor 1
Q13177	PAK2	PAK-2/p58	Serine/threonine-protein kinase PAK 2
P26678	PLN	PLB	Cardiac phospholamban
P43088	PTGFR	PGF2α-R	Prostaglandin F2-alpha receptor
Q13464	ROCK1	ROCK1	Rho-associated protein kinase 1
O75116	ROCK2	ROCK2	Rho-associated protein kinase 2
Q92736	RYR2	RYR2	Ryanodine receptor 2
P00441	SOD1	Sod1	Superoxide dismutase [Cu-Zn]
Q13586	STIM1	STIM1	Stromal interaction molecule 1
P21731	TBXA2R	TXA_2_-R	Thromboxane A2 receptor
A0A499FJK2	TGFB1	TGFβ1	Transforming growth factor beta
P37173	TGFBR2	TGFR-2	TGF-beta receptor type-2
P21980	TGM2	TGM2	Protein-glutamine gamma-glutamyltransferase 2 (Transglutaminase 2)
P07996	THBS1	TSP-1	Thrombospondin-1
P01375	TNF	TNFα	Tumor necrosis factor
Q9Y210	TRPC6	TrpC6	Short transient receptor potential channel 6
Q8TD43	TRPM4	TrpM4	Transient receptor potential cation channel subfamily M member 4
Q9H4B7	TUBB1	Tubulin β1	Tubulin beta-1 chain
Q9GZT5	WNT10A	Wnt-10a	Protein Wnt-10a
P56704	WNT3A	Wnt-3a	Protein Wnt-3a
P41221	WNT5A	Wnt-5a	Protein Wnt-5a
Q9NUD5	ZCCHC3	ZCHC3	Zinc finger CCHC domain-containing protein 3

**Table 2 ijms-25-02601-t002:** List of significantly most-enriched KEGG pathways (ranked according to *p*-value).

Pathway Term	% Occurrence *	*p*-Value	FDR **
Calcium signaling pathway	26	2.80 × 10^−9^	2.80 × 10^−7^
Proteoglycans in cancer	24	3.90 × 10^−9^	2.80 × 10^−7^
cGMP-PKG signaling pathway	20	1.20 × 10^−7^	5.60 × 10^−6^
AGE-RAGE signaling pathway in diabetic complications	16	6.10 × 10^−7^	1.80 × 10^−5^
Diabetic cardiomyopathy	20	6.30 × 10^−7^	1.80 × 10^−5^
Pathways in cancer	28	1.40 × 10^−6^	3.20 × 10^−5^
Platelet activation	16	2.60 × 10^−6^	5.20 × 10^−5^
Renin secretion	12	2.40 × 10^−5^	4.20 × 10^−4^
Relaxin signaling pathway	14	4.40 × 10^−5^	6.80 × 10^−4^
Pathways of neurodegeneration—multiple diseases	22	1.10 × 10^−4^	1.50 × 10^−3^
Cushing syndrome	14	1.20 × 10^−4^	1.50 × 10^−3^
Dilated cardiomyopathy	12	1.60 × 10^−4^	1.90 × 10^−3^
Focal adhesion	14	5.20 × 10^−4^	5.60 × 10^−3^
Alzheimer disease	18	6.10 × 10^−4^	6.10 × 10^−3^
Oxytocin signaling pathway	12	1.00 × 10^−3^	9.70 × 10^−3^

*: % of age-dependent proteins occurring in the respective pathways. **: False Discovery Rate.

**Table 3 ijms-25-02601-t003:** List of significantly most enriched REACTOME pathways (ranked according to *p*-value).

Pathway Term	% Occurrence *	*p*-Value	FDR **
Muscle contraction	5	1.40 × 10^−7^	5.81 × 10^−5^
EPH-Ephrin signaling	8	6.89 × 10^−7^	1.43 × 10^−4^
Extra-nuclear estrogen signaling	8	4.91 × 10^−6^	6.77 × 10^−4^
Signal Transduction	1	7.76 × 10^−6^	8.07 × 10^−4^
Ion homeostasis	9	1.17 × 10^−5^	9.67 × 10^−4^
Signaling by GPCR	2	2.06 × 10^−5^	0.001
GPCR ligand binding	2	3.82 × 10^−5^	0.002
Signal amplification	11	4.11 × 10^−5^	0.002
Cardiac conduction	5	7.34 × 10^−5^	0.003
Smooth Muscle Contraction	9	9.89 × 10^−5^	0.004
Nuclear Receptor transcription pathway	6	1.05 × 10^−4^	0.004
Hemostasis	2	1.11 × 10^−4^	0.004
Platelet homeostasis	6	1.16 × 10^−4^	0.004
Signaling by Receptor Tyrosine Kinases	2	1.57 × 10^−4^	0.005
Axon guidance	2	1.75 × 10^−4^	0.005

*: % of age-dependent proteins occurring in the respective pathways. **: False Discovery Rate.

**Table 4 ijms-25-02601-t004:** Predicted Functional Partners with proteins in STRING network (very high confidence).

Gene Name (Homo Sapiens)	Protein Name	Score
BCL3	B-cell lymphoma 3 protein	0.999
COX6A1	Cytochrome c oxidase subunit 6A1, mitochondrial	0.999
DCN	Decorin	0.999
NFKBIA	NF-kappa-B inhibitor alpha	0.999
PFN1	Profilin-1	0.999
RELB	Transcription factor RelB	0.999
TGFB3	Transforming growth factor beta-3 proprotein	0.999
TIMP1	Metalloproteinase inhibitor 1	0.999
TNFRSF1A	Tumor necrosis factor receptor superfamily member 1A, membrane form	0.999
UQCRC1	Cytochrome b-c1 complex subunit 1, mitochondrial	0.999

## Data Availability

Not applicable.

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
