# Peer review of "Functional, Structural and Proteomic Effects of Ageing in Resistance Arteries"

_ijms, 2024, doi:10.3390/ijms25052601_

Round 1

Reviewer 1 Report

Comments and Suggestions for Authors

This is a very informative and well-written review. i have a few suggestions:

- At p. 3 it would be informative to shortly provide some detail about the type of microcirulatory damage that is casued by the pressure wave. For example, leakage, faulty angiogenesis, inflammatory foci etc. More general, the things that logically connects the pressure and the damage.

- p. 6: "Although the Cav3.2 ... pathophysiological role in age-related vascular dysfunction." Since mature and middle aged rodents are discussed in this sentence, it is not per se clear that this part is discussing a pathological remodeling. It might still be physiological. Perhaps some explanation is needed.

- at p 8, the part about FMVD. it might be worthwhile to inlcude the interesting work by S. Segal about the age-induced shift from NO toward mitochondrial Ca2+-induced EDHF. i would also recommend to explain EDHF shortly at the start of this section, and the well-known interchange between NO and EDHF that is often seen (although much is from observations in eNOS KO mice), and the role of H2O2 as an EDHF.

Author Response

The following are my response to the reviewer’s comments:

- Reviewer 1

- At p. 3 it would be informative to shortly provide some detail about the type of microcirculatory damage that is caused by the pressure wave. For example, leakage, faulty angiogenesis, inflammatory foci etc. More general, the things that logically connects the pressure and the damage.

Reply: Thank you for the suggestion. I have included one sentence on Page 4 stating: “The types of microcirculatory damage incurred by the increased pressure pulsatility include decreased angiogenesis, capillary rarefaction, increased inflammation, BBB rupture, microhemorrhages, and in case of the brain, white matter lesions [9,10].”

 - p. 6: "Although the Cav3.2 ... pathophysiological role in age-related vascular dysfunction." Since mature and middle aged rodents are discussed in this sentence, it is not per se clear that this part is discussing a pathological remodeling. It might still be physiological. Perhaps some explanation is needed.

Reply: Thank you for the suggestion. I have included additional text in that sentence to improve clarity: “Although the CaV3.2 expression in wildtype mice was unaffected by ageing, we found that this protective role of the CaV3.2 channels in young animals against excessive myogenic tone vanished in mature adult and middle-aged mice, suggesting that a loss of CaV3.2 channel function may play a pathophysiological role in age-related vascular dysfunction[47]”. Please let me know if the subject still need clarification.

- at p 8, the part about FMVD. It might be worthwhile to include the interesting work by S. Segal about the age-induced shift from NO toward mitochondrial Ca2+-induced EDHF.

Reply: Thank you for the suggestion. I have added text on Page 9 as follows: “In superior epigastric arteries from old mice, the age-dependent decline in NO-mediated vasodilation was compensated for by an increase in mitochondrial Ca2+-release and activation of SK/IK Ca2+-activated K+ channels acting as EDH [84]. Thus, the endothelium may in skeletal muscle arteries be resilient to age-induced loss of NO production through the upregulation of EDH”.

I would also recommend to explain EDHF shortly at the start of this section, and the well-known interchange between NO and EDHF that is often seen (although much is from observations in eNOS KO mice), and the role of H2O2 as an EDHF.

Reply: Thank you for the suggestion. I have added the following text also on Page 9: “H2O2 released from the endothelium acts as a vasodilator via activation of smooth muscle K+ channels, thereby eliciting Endothelium-Dependent Hyperpolarization (EDH) and -vasodilation [76,77]. EDH is defined as the ability of an endothelium-dependent vasodilator to cause dilation via hyperpolarization of smooth muscle cells derived by a factor released from the adjacent endothelial cells (K+; H2O2; endothelial hyperpolarization spreading to smooth muscle via myoendothelial junctions) [78]. In resistance arteries, it is widely accepted that EDH functions as a compensating vasodilator mechanism when NO-mediated vasodilation is reduced [79]”.

Reviewer 2 Report

Comments and Suggestions for Authors

I have read with great interest and attention the paper titled 'Functional, Structural and Proteomic Effects of Ageing in Resistance Arteries.' Although I believe the author has conducted an extensive literature review, I must admit that the paper, in its current form, does not seem suitable for potential publication. Essentially, the major problem with the work is that it is a (too) old-style narrative review: the entire text might be more suitable for a monograph or a textbook.

I would suggest to the author to completely reorganize his/her paper by moving all the details regarding the literature review process (time frame, consulted databases, keywords, types of included works, evaluated outcomes) into the methods section. After doing so, the results section should present (in a significantly more concise manner than the current paper) the obtained results.

Author Response

- Reviewer 2

Although I believe the author has conducted an extensive literature review (Author reply: Thank you!), I must admit that the paper, in its current form, does not seem suitable for potential publication. Essentially, the major problem with the work is that it is a (too) old-style narrative review: the entire text might be more suitable for a monograph or a textbook.

Reply: Thank you for the criticism, which I do not fully agree with. When I look at the call for this special collection of papers in IJMS, I do not see that a review may not be structured in the format of a “systematic literature review” as I have done. I aimed to clarify the impact of ageing on resistance artery tone regulation, particularly for the three important types that I am interested in: myogenic tone, flow-mediated vasodilation, and structural remodelling. I have often encountered misleading or biased representations of the impact of ageing in the vasculature, and I figured that the only way to suppress this would be to review the field in the most comprehensive manner, and by precluding the author from introducing new bias or by predominantly citing papers agreeing with his/her own previous work. I admit that parts of my literature overview might be too dense with information from the various studies cited, but I have tried to overcome this in the revised version by using a clearer and more transparent structure, which is outlined under a new heading titled: “Review Methodology and Structure”.

The text of this special call furthermore states that:

 “The special issue aims to introduce current knowledge, frontier technologies and multidisciplinary applications on arterial ageing. Topics include: signaling pathways and mechanisms underlying the senescence of endothelial cells; structural and functional abnormalities of aged artery; cross-talk between different types of cells within the arterial wall; strategies for preventing and reverting endothelial senescence and arterial ageing. “

In my review, besides the systematic literature approach, I have paid special attention to include frontier technologies (Mass-Spec-based proteomics, and Bioinformatics); structural/functional abnormalities in aged arteries; and cross-talk between different cell types. Besides, Tables 1-4 and Fig. 1 provide a comprehensive list of proteins and signalling pathways involved in ageing of the vasculature. Finally, Fig. 2 summarizes the functional/structural changes in simple schematic form, and briefly points out which molecular changes and signalling pathways are involved in the respective processes.

I would suggest to the author to completely reorganize his/her paper by moving all the details regarding the literature review process (time frame, consulted databases, keywords, types of included works, evaluated outcomes) into the methods section.

Reply. Thank you for the suggestion.  I have done this by adding a new section under the title “Review Methodology and Structure”, see page 3.

After doing so, the results section should present (in a significantly more concise manner than the current paper) the obtained results.

Reply. Thank you for the suggestion.  The structure and readability was improved by making small explanatory text insertions highlighted to divide the text up in sub-sections. In some cases, the order of papers/information cited was changed, to accommodate a better structure and clarity. I have meticulously, scrutinized the text to include/exclude words that might improve/disturb the reading experience. I hope reviewer 2 when he/she reads the entire text now feels that the reviewed information is presented in a significantly more concise manner.

Round 2

Reviewer 2 Report

Comments and Suggestions for Authors

I would congratulate the Authors for his huge work in revising the paper

The whole document, however, is necessary too long and verbose

As I already suggested, we need a much (much) shorter paper 

Author Response

Reply: Thank you for the comments. I have again meticulously removed all unnecessary words or phrases, as well as condensed the sections on age-dependent changes in myogenic tone, FMVD and structural remodeling. One reference was removed due to redundancy (from 136 to 135 references). Naturally, I could not condense the figures or tables, as they contain vital physiological and molecular information pertaining to the aim of this review as well as to the scope of this journal.

The text in the original submission was 12855 words long, and the revised version 1 was 13373 words long. This increase in length was unfortunate given the previous request of Reviewer 2, but was deemed necessary since reviewer 1 requested new information regarding EDH being added to the manuscript. The resubmitted version 2 now has 12655 words and is one page shorter than the first revision. Thus, text has been reduced at the request of Reviewer 2 in the first and second revised versions (although more text was also added in Rev. 1). I have now reached the limit to how much I can reduce the length of the manuscript without compromising the idea behind the review and the journal call. If more information was to be cut, I would have to delete whole sections in the paper, which would damage the structure and the overall scientific content of my review. I believe, as stated previously, that there is a need to present the data on the effects of ageing on resistance arteries in a systematic and comprehensive manner, which means that all relevant studies and data need to be included. I do hope that Reviewer 2 acknowledges this, as well as my large efforts to meet his request.